# Aging in Place and Healthcare Equity: Using Community Partnerships to Influence Health Outcomes

**DOI:** 10.3390/healthcare13101132

**Published:** 2025-05-13

**Authors:** Annie Rhodes, Christine C. McNichols

**Affiliations:** 1Department of Gerontology and Virginia Center on Aging, College of Health Professions, Virginia Commonwealth University, Richmond, VA 232298, USA; mcnicholsc@vcu.edu; 2Department of Occupational Therapy, College of Health Professions, Virginia Commonwealth University, Richmond, VA 23298, USA

**Keywords:** aging in place, health disparities, health equity, community-based organizations

## Abstract

**Background and Objective**: Aging in place (AIP) refers to the ability to remain in one’s home and community as one ages. While AIP is widely regarded as beneficial, disparities in housing stability, accessibility, and affordability create inequitable barriers. Current clinical AIP interventions focus on individual-level solutions, often overlooking broader socio-economic and structural determinants.This study examines how community-based interventions, particularly those from Rebuilding Together Richmond (RT-R), address these gaps through home modifications and critical repairs. **Methods**: Using the National Institute on Minority Health and Health Disparities (NIMHD) Research Framework, demographic and service data from home modifications from a community-based organization, RT-R was analyzed. Descriptive statistics assessed the characteristics of homeowners served, the types of repairs performed, and their potential impact on AIP. Repairs were categorized as structural or occupational to evaluate their contributions to home safety and accessibility. **Results**: RT-R provided repairs for 33 homes, benefiting 47 individuals all of whom were Black or African American living in a ZIP code with high eviction rates. The majority (63.8%) were female, and 51% were older adults and/or had a disability. Structural repairs were more frequent than occupational modifications reflecting both homeowner needs, service availability, and community organizational goals. **Conclusions**: Housing stability is a critical yet overlooked factor in AIP. Integrating clinical AIP interventions with community-based solutions can more effectively address health disparities, reduce institutionalization risks, and improve long-term livability. Partnerships between healthcare practitioners and organizations like Rebuilding Together are essential to advancing equity in AIP. Access to housing is not *accessible* housing, and to remove barriers, practitioners and community-based organizations should expand their appreciation of obstacles to include historical, contemporary, economic, and environmental factors to work toward equity in aging in place for all.

## 1. Introduction

### 1.1. What Is Aging in Place, and Where Are the Current Gaps?

As defined by the National Institutes of Aging (NIA), the concept of aging in place (AIP) means “staying in your own home as you get older” [1]. Over the last 60 years, the use of the term has steadily increased in academic literature, and the concept of supporting individuals’ AIP has moved into clinical practice, wherein healthcare professionals help older adults to AIP [2,3]. AIP (more specifically, not living in institutional environments) has numerous associated benefits, such as protecting against healthcare-associated infections, extending lifespan, and honoring individual preferences [4,5]. Despite its benefits, the critical literature has highlighted that the ability to AIP is intersectional and influenced by individual, interpersonal, community, and societal factors [4,6]. These factors affect many domains across the lifespan, including individual health, relationships in the built environment, and socioeconomic status [7]. Despite the complex multi-level factors mediating AIP, clinical interventions primarily focus on the individual level and physical environment. A lack of awareness of the other domains of influence and factors influencing AIP contributes to persistent disparities in who “gets” to AIP and limits the practitioner’s ability to provide assistance and interventions to support AIP.

This study bridges the gap between theoretical aging-in-place literature and the clinical practice of mediating aging-in-place barriers through a collaboration with the organization, Rebuilding Together. Rebuilding Together (RT-N), a nonprofit organization that performs home repairs for individuals in need, allowed for a unique understanding through the use of descriptive analyses of the individual needs to enable sustained livelihood in home environments. This study uses data from the Richmond chapter of Rebuilding Together (RT-R). Our primary aims are (1) to review AIP ecology and associated disparities using a critical lens via the Health Disparities Research Framework, (2) to present the results of a national community-based organization (CBO) and data that mediate barriers to AIP, and (3) discuss the implications for how clinical AIP interventions and supports should look to support equity-based approaches to AIP.

### 1.2. The Ecology of AIP

The National Institute of Minority Health and Health Disparities Research Framework (NIMHD) (Figure 1) displays the intersection of levels of influence and domains of influence over the life course with health outcomes for individual health and family/organizational, population, and community health [7]. This framework was used to define each domain of AIP, and then interpreted using a critical gerontological lens (Table 1) [8,9,10]. The purpose of the critical lens was to identify structural conditions contributing to unequal experiences where factors beyond personal choice significantly influence AIP. From a clinical stance, analyzing and integrating all facets impacting AIP facilitates an improved understanding of barriers to AIP and, ultimately, the development of interventions that remove these barriers.

### 1.3. Access to Safe, Livable Housing Is Not Evenly Distributed

Physical environments and healthcare systems are two domains influencing AIP (see Figure 1). Household environments, a type of physical environment, are one component of AIP ecology. In the United States, only 10% of homes are considered “aging ready” [11]. A person’s household is based upon personal values, resources, and familial structure and is individually tailored to meet their needs. However, as individuals age or encounter a new health incident, the needs of the household environment may immediately change without the resources or time available in order to adjust the physical environment to meet new demands.

At the individual level, homeownership is closely linked to economic security and greater confidence in the ability to AIP. Individuals with stable housing and access to healthcare are better positioned to AIP successfully [12]. At the societal level, economic insecurity—including housing precarity—disproportionately affects racial and ethnic minorities and people with disabilities across the lifespan [13]. Disability status and poverty are strongly correlated, with adults with disabilities being 2.5 times more likely to live in poverty than those without disabilities; the poverty rate among adults with disabilities is 27%, compared to 12% among adults without disabilities. Nearly a quarter of non-Hispanic White Americans with disabilities live in poverty (24%), compared to nearly 40% of African Americans with disabilities [14]. The relationship between disability, poverty and age is intersectional and includes the increasing risk of disability with old age, and increased difficulty finding employment for older adults and people with disabilities [14,15].

Inadequate and inaccessible housing stock exacerbates these disparities. A 2021 analysis found that 19% of U.S. households have an accessibility need, largely because housing is typically designed for able-bodied individuals [16,17]. Accessibility needs are disproportionately concentrated among low-income households, racial and ethnic minorities, veterans, older adults, and renters [14]. These groups also face reduced access to livable communities—safe and secure environments with supportive features and services—or they are more cost-burdened in securing housing [6,18].

Clinical and practical interventions that improve physical housing, such as modifications and safety assessments, play a significant role in supporting individual health [19]. However, these interventions primarily operate at the personal level and do not address broader community or societal impacts. Occupational therapists (OTs), are a part of the healthcare system, covered by health insurance, and can be one of the primary providers of clinical AIP support.

Occupational therapy AIP practice focuses on individual and interpersonal levels of influence, enhancing home safety and quality of life to enable continued participation in valued activities [20]. OT services include education, tools, and recommendations for home modifications that help older adults remain in their homes [21]. Healthcare interventions, such as OT services, have numerous positive outcomes. Home evaluations and modifications can reduce falls and injuries, decrease fear of falling, and improve overall home safety [19]. Prior research has demonstrated that home modifications such as grab bars and shower chairs decrease the risk of nursing home admission. A recent systematic review of home modifications found that in twelve randomized controlled trials of home modifications for older adults, those that incorporated OT-driven home modifications in combination with other clinical and behavioral components, or physical activity interventions had positive outcomes [22]. Access to OT assessments, as well as the implementation and initiation of these services can be a barrier. Traditional Medicare will cover the cost of home assessment, which an OT could perform; however, home modifications are not a covered service. Medicare Advantage plans and Medicaid may offer minimal coverage. However, there are inconsistencies across plans and states [23]. Consequently, the lack of financial resources and systemic inequities often prevent OT practitioners from completing in-home modifications.

When provided, clinical interventions are impactful. However, the provision of these interventions relies on the recipient having stable shelter, agency over that shelter to make modifications, and the financial means to afford these services [2,3,24,25]. In the Black community, the homeownership rate is 46.4%, compared to 75.8% in the White community in the United States [26]. Relatedly, American Black families have a median wealth of approximately USD 17,600 and approximately 10% of the median wealth of White families (USD 171,000) [27,28,29]. Housing stability is a critical determinant of health, yet it is disproportionately inaccessible to non-White and/or disabled individuals. Financial and structural barriers—including poverty and inaccessible housing—are at a heightened risk of housing instability, threatening the ability to AIP. The promotion of high-quality livable homeownership across the lifespan may help to prevent institutionalization in late life.

### 1.4. A Community-Based Organization Intervention and Mediating Barriers to AIP

Using community–clinical partnerships to increase access to home modifications, such as those provided by OTs, can support health equity. Limited research has focused on the integration of community and healthcare partnerships to remove barriers to AIP; however, the existing evidence is promising. For instance, Grasso et al. (2023) [30] incorporated two OT home modification visits in a quasi-experimental study with the national organization Rebuilding Together (RT-N). Participants had statistically significant increases in their performance and satisfaction (as measured by the Canadian Occupational Performance Measure), and the average total home modification costs were over USD 10,000.00 in savings to the homeowners [30]. Having OTs provide home modifications executed by a community partner, such as RT-N, removes community and societal barriers associated with clinical AIP interventions, such as insurance coverage and cash needed for repairs.

Rebuilding Together (RT-N) is a national nonprofit organization that performs essential repairs to allow individuals to remain in their homes. The organization seeks to make meaningful changes and measurable impacts through providing services and has assisted over six million people in the last 30 years since its inception. The foci of the organization are multifold, and it seeks to make critical home repairs for low-income owners. RT-N reduces individual cost burden and can even demonstrate positive community health outcomes such as increased livability and improved home values. RT-N efforts benefit many groups, including veterans, individuals with disabilities, and communities impacted by natural disasters. Repairs have mutually inclusive benefits, including helping older adults AIP, assisting families to preserve generational wealth, and preventing homelessness [31]. Unlike healthcare practitioners, who are often limited to working at an individual level of influence, RT-N is able to provide repairs that support interpersonal, community, and societal influence, including improving neighborhood livability, providing repairs unrelated to a health issue, and supporting multiple generations in the same home.

## 2. Materials and Methods

To examine the AIP ecology across all levels of influence and study the structural impact and outcomes for one community, retrospective secondary data from Rebuilding Together Richmond (RT-R) were analyzed. Housing precarity and economic security are common in Richmond Virginia, which has an overall eviction rate of 11%, the second highest in the United States [32]. Certain low-income neighborhoods experience rates 2–3× higher. In 2022, RT-R focused all repair activities on a single ZIP code—23224. In ZIP code 23224, 12%–20% of renters will face eviction—one of the highest eviction rates in Richmond [32,33,34]. After receiving institutional review board approval from Virginia Commonwealth University (located in Richmond and the authors’ home institution), data extraction occurred without any homeowner identifiers. Information technology team members at Rebuilding Together supported data extraction and preservation through FileLocker, a secure file-sharing service in a single file. The study was conducted between October 2024 and March 2025. The data were cleaned and analyses performed in R Studio Version 2023, 12.1 for demographics and frequencies.

RT-R uses a 22-category system to organize repairs: attic, bathroom, bedroom, crawl, dining, electrical, external door, fence, gutters, hall, HVAC, kitchen, landscape, and miscellaneous—paint, pests, plumbing, porch/deck, ramp, roof, siding, windows. For analysis, repairs were classified using the RT-R categorization data dictionary (provided to the research team by RT-R), with two research team members analyzing and verifying each repair. Homeowners were categorized into four groups: Aging with a disability (65+ and a disability), Aging in Place (65+ and no disability), Living with a disability (Under 65 with a disability), and All others. To further delineate the classification of home repairs, each individual repair was classified as either “structural” or “occupational” based on dual reviewer alignment, with most homes receiving multiple repairs [35]. The American Occupational Therapy Association (AOTA) Occupational Therapy Practice Framework served as a basis for item categorization and determination (Table 2 and Table 3). Structural repairs were classified as repairs that addressed the home’s safety and/or aesthetics. Some examples of these repairs included the replacement of the outlet cover, new paint, and replacement of the window(s). Occupational repairs were individually classified based on the potential impact on a resident’s functional mobility, activities of daily living (ADLs), and/or instrumental activities of daily living (IADLs). Some examples of occupational repairs included the addition and/or replacement of handrails for stairs, raising toilet heights, and the addition of grab bars within the shower, amongst other home repairs. This separation allowed for the delineation of a deficit or disability based modification from modifications or repairs which improved home value, comfort, livability, or safety [35].

## 3. Results

Rebuilding Together Richmond served 47 people in 33 homes in the Richmond ZIP code 23224 in 2022 (Table 4). Of these 47 individuals, 100% were Black or African American, and the mean age was 65.76 years. Twenty of the homes were deemed in need of critical repairs (62.5%), 30 (63.8%) recipients were female, and 17 (36.2%) were male. The average income was USD 29,313.85, ranging from USD 9,252 to USD 53,940.48. The median income was USD 31,190.04. All homeowners and family members were within <30% of the area median income (AMI) (USD 62,671) [35]. Five recipients were veterans, twelve individuals identified as having a disability, twenty-four individuals were older adults (age 65+), and seven were both older adults and had a disability.

An average of 2.7 structural repairs and 0.94 occupational repairs were performed per household in the group of older adults without a disability. Most commonly, this group needed repairs in the porch/deck (occupational), with 52.9% of recipients needing a porch/deck repair. For those aging with a disability, the most common repair need was in the bathroom (occupational). Of those aging with a disability, 42.9% needed a bathroom repair. For those under 65 with a disability, the most common repair needed was crawl space—60%. Porch/deck was the most common repair needed for those without a disability, and for those under 65, 50% received a porch/deck repair.

## 4. Discussion

In analyzing the differences between structural and occupational repairs from the data, structural repairs were performed at a higher frequency than occupational repairs. This was to be expected, as RT-R performs repairs from a physical and structural-based origin. Home modifications, while an important skill for OT practitioners, can encompass remodeling physical structures (such as what was performed in this study). Additionally, it can include item adjustments (such as repositioning), replacement of objects (requiring less physical force/effort such as different levels/handles), as well as the removal of objects (barriers to safety or livability). As such, much of the value that OTs and other healthcare specialists can provide with the alteration of home environments would be difficult to provide solely by a CBO. Conversely, many, if not all of the structural repairs, would not be covered by medical insurance because they are not directly deficit-based. Structural or occupational modifications can make large-scale livability differences for individuals in their homes and their neighborhood. Structural repairs also support home value and aesthetics, which supports intergenerational wealth building. As such, we sought to analyze current CBO service provision, as well as how further opportunities may exist for CBO organizations and rehabilitation providers.

In this study, bathrooms had the highest frequencies of repairs in adults with a disability. Bathrooms are a high-risk area for falls, particularly due to water (after bathing/showering), as well as the challenges of entering/exiting tub showers and thresholds, which pose particular problems for safe functional mobility while performing ADLs. The installation of grab bars, conversion of washing thresholds, and elevation of toilet heights would be areas of high impact to individual homeowners and were some of the repairs performed in bathroom environments by RT-R. Based on previous studies, these modifications are likely to protect against falls and ensure enhanced livability and quality of life. Previous analyses of Medicare beneficiaries have demonstrated that bathroom modifications prevent repeated falls, and that racial and ethnic minorities are less likely to have bathroom modifications, leading to subsequent falls [36].

### 4.1. Implications

There are several implications of this study. The most significant one is that the concept of the “AIP” paradigm, which primarily focuses on the physical structure of the home, may unintentionally be exclusionary to diverse populations. AIP, as it is defined by the National Institute of Aging and the AARP, does not provide options for individuals who do not have agency over their living domain, nor does it address livability factors which are relevant to the immediate environment outside the dwelling. In the context of health, community and social factors have significant implications and have historically been inequitable for racial and ethnic minorities.

From an international standpoint, the majority of AIP literature is based in the United States, or western Europe, highlighting a gap. Audiences outside of these countries should study and disseminate culturally relevant definitions of AIP if they already exist; if a country does not currently have infrastructure or a paradigm for AIP, they could consider building infrastructure which includes all levels of health and look for more comprehensive ways to define and share the best practices for AIP.

Prior research has been conducted with community-based research partnerships (CBRP) in order to achieve health equity. Ward and colleagues (2018) proposed a Conceptual Model for Evaluating Equity within the Context of CBPR Partnerships framework, suggesting that there are various long-term outcome measures that result from CBRP effectiveness in creating equity, which include improvements in social and environmental conditions within inequitable communities [37]. Additionally, improvements within individuals’ physical, mental, and social health could be addressed within communities which have health inequities with the use of CBRPs. Limited research literature to date has addressed research participation with community-based organizations; however, this study provides early foundational knowledge of some of the manners in which outputs from a CBRP could add individual and community-level value to better address rectifying health inequities.

### 4.2. Limitations

There are several limitations to the study. RT-R collects information relevant to the domain of the physical and built environment; data points relevant to the health domain (such as improvements in IADLs or reductions in falls) are not collected by the RT-R program. This study is cross sectional, not longitudinal, and therefore we are unable to gauge the long-term impact of the modifications. It is likely that repairs such as installing toilets and repairing hand railings have supported AIP for participants.

Analytically, statistical tests appropriate to this data were limited. Using ANOVA to detect the significance between categories of homeowners was not appropriate due to clustering (instances of multiple occupants in one house). Analysis comparing different types of repairs at the home level was also not appropriate, because homes received multiple repairs in multiple categories, violating the independence of assumptions. The small sample size (n = 33 homes, and n = 47 participants) made a mixed-effect regression inappropriate, due to being underpowered. All data were collected in a single ZIP code, making the generalizability of the findings limited.

### 4.3. Future Directions

The limitations of the data provide a backdrop for the discussion of (re)conceptualizing AIP. From an implementation standpoint, clinical interventions have primarily focused on AIP as the intersection of physical/built environment and the individual and interpersonal levels. However, AIP includes community levels (such as neighborhood livability) and societal domains (such as healthcare policy, discrimination, and social norms). RT-R concentrates their work in particular neighborhoods to support individual health and as a way to improve neighborhood livability. Partnership with CBOs and healthcare may more effectively address AIP by impacting more levels of influence. Proactive partnerships between CBOs and healthcare organizations also create the potential for more AIP research, because outcomes can be robustly linked to health outcomes. Future studies could follow participants longitudinally to gauge effectiveness on the individual, interpersonal, community, and societal levels.

Occupational therapists, aging-in-place specialists, and other clinical providers and interventions can be co-designed with community partners to minimize challenges in completing ADLs and IADLs and to support long-term goals. These goals can support community health, such as intergenerational home ownership, neighborhood livability, and reduce the burden of cash repairs. It also provides an avenue for a new conceptualization of the idea of AIP in the clinical space.

This study also serves as a call to action for healthcare practitioners and educators to change how we define and support AIP. Focusing on the physical and built environment at the individual levels (i.e., “providing [ing] tools to allow older adults to stay in their homes as they age by supporting the necessary modifications”) [21] entrenches inequities by not addressing community and societal barriers.

## 5. Conclusions

AIP is a multilevel, lifespan process. Barriers can occur at any level, but the resolution of these barriers supports person-centered practice. A total of 77% of older adults would prefer to AIP [38]. Healthcare practitioners are trained to resolve disability-based barriers by modifying the built environment. While impactful, this approach is limited in several ways and makes implicit assumptions. It assumes that barriers to AIP are a result of disability and can be resolved by managing that disability. It also assumes that the recipient has stable shelter, the agency over that shelter to authorize modifications, and the financial means to afford modifications. The majority of RT-R repairs occurred in the structural category. A purely clinical intervention may not have performed these repairs. The implications of these assumptions should be appreciated in terms of the broader socio-cultural influence. Social norms such as discrimination and the consequences of historical and contemporary policies manifest in adverse outcomes in home ownership, wealth, and ability. Racial and ethnic minorities and people with disabilities are less likely to benefit from the current AIP approach because they are less likely to own a home and, on average, have much fewer financial resources.

By analyzing outputs from a single CBO, RT-R, we present a possibility of expanding access to AIP, and suggest a paradigm shift. Moving away from the idea of deficit-focused built environment modifications and into housing stability, neighborhood livability, and intergenerational homeownership.

Access to housing must be reframed as access to *accessible*, *affordable*, and *equitable* housing, requiring practitioners and community organizations to address historical, economic, and environmental barriers. Partnerships between healthcare providers and community-based organizations offer a scalable and necessary model for dismantling inequities that traditional, individually focused clinical interventions do not address. Future work should build on these insights by developing and evaluating co-designed interventions that promote not only individual well-being but also strengthen community health and resilience across the lifespan.

## Figures and Tables

**Figure 1 healthcare-13-01132-f001:**
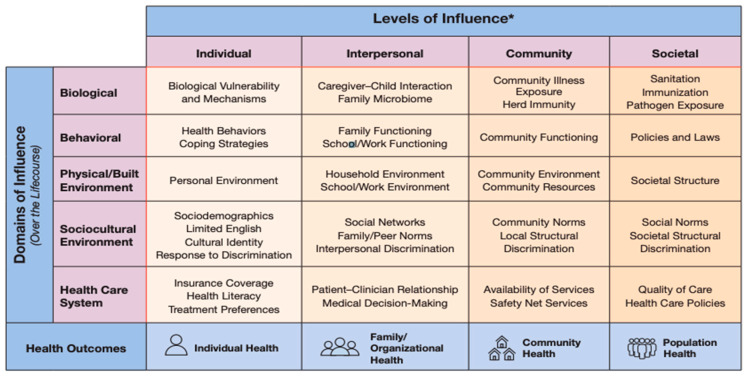
National Institute on Minority Health Disparities Research Framework [7]. * NIH-designated Populations with Health Disparities: Racial and Ethnic Minority Groups (defined by OMB Directive 15), People with lower socioeconomic Status, Underserved Rural Populations, People with Disabilities, Sexual Minority Populations.

**Table 1 healthcare-13-01132-t001:** Application of National Institute of Minority Health and Health Disparities Research Framework on the influences on clinical and community aging in place.

Level of Influence	Clinical Aging in Place Intervention	Rebuilding Together Intervention
Individual	Provide advice and assessment on mobility (deficit focused)	Not fully addressed
Interpersonal	Provides modifications related to health condition or safety in health concern (deficit focused)	Provide modification or repairs not directly associated with health condition/enhancement (environmental). Not contingent on health condition/not limited to deficits.
Community	Not addressed	Relieve the economic burden by providing no-cost repairs. Expand the availability of repairs, combatting displacement/reducing displacement/institutionalization.
Societal	Not addressed	Improve neighborhood livability, improve home values in census blocks. Expand AIP to new populations.

**Table 2 healthcare-13-01132-t002:** Occupational repairs performed.

	Attic	Bathroom	Hall	Kitchen	Porch/Deck Railway
Aging in Place Over 64	0	3	1	3	9
Aging with a Disability Over 64	0	3	0	0	0
Living with a Disability Under 65	0	1	0	2	1
All Others	1	1	0	0	9

**Table 3 healthcare-13-01132-t003:** Structural repairs performed.

	Bed Room	Crawl	Dining Room	Electrical	EXT Door	Fence	Gutters	HVAC	Landscape	Paint	Pests	MISC	Ramp	Roof
Aging in Place Over 64	0	4	0	6	6	3	4	2	2	4	1	3	0	6
Aging with a Disability Over 64	0	0	0	0	0	0	1	0	0	0	0	0	0	1
Living with a Disability Under 65	1	3	1	1	1	0	0	1	1	2	0	0	1	1
All Others	1	3	0	4	3	6	4	2	4	2	2	0	0	1

**Table 4 healthcare-13-01132-t004:** Categories of repairs performed by Rebuilding Together.

	Mean Structural	Total Structural	Mean Occupational	Total Occupational	N (%)
Aging in Place Over 64	2.76	47	0.94	16	17 (36.17)
Aging with a Disability Over 64	0.42	3	0.42	3	7 (14.89)
Living with a Disability Under 65	2.8	14	0.8	4	5 (10.63)
All Others	2.05	37	0.61	11	18 (38.29)
Total	2.14	101	0.72	34	47

## Data Availability

The datasets presented in this article are not readily available. Datasets may include identifiers and access to the data is regulated by the IRB. Questions about the datasets should be directed to Annie Rhodes, PhD at rhodesas2@vcu.edu.

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
