# Peer review of "Aging in Place and Healthcare Equity: Using Community Partnerships to Influence Health Outcomes"

_healthcare, 2025, doi:10.3390/healthcare13101132_

Round 1
Reviewer 1 Report
Comments and Suggestions for Authors
- Please reformat Abstract according to the journal format
- Please add date of the study
- Please amend: Data was to data were
- Please provide justification how international audiences may be benefit the findings
- Please provide about IADL before and after interventions
- Discussion section has not well developed. It should be supported with related theories and literature
- Please provide limitations of the study
- Please provide implications of the study
Author Response
Reviewer 1
Reviewer Comment |
Author Response |
1. Please reformat Abstract according to the journal format
|
We adjusted Background/Objective to the heading Background and Objective. All other components of the abstract have remained the same.
|
2. Please add date of the study
|
We thank the reviewer for the comment, we have added dates of the study in the Methodology section
|
3. Please amend: Data was to data were |
We thank the reviewer for that comment, all instances using “data was” have been updated to “Data were”
|
4. Please provide justification how international audiences may be benefit the findings
|
Thank you reviewer we added a sentences in “Implications” which speak to this specifically From an international standpoint, the majority of AIP literature is based in the United States, or western Europe. This is a gap. Audiences outside of these countries should study and disseminate culturally relevant definitions of AIP if they already exist, if a country does not currently have infrastructure or a paradigm for AIP, The implications of this article are that the definitions have been to narrow, , they should be mindful of the gaps in the current paradigm and look for more comprehensive ways to define, and share best practices on AIP
|
5. Please provide about IADL before and after interventions |
We thank this reviewer for the comment, unfortunately our study is cross sectional and not longitudinal. We have added a note in the methodology section and the limitations section about how future studies should follow participants longitudinally
|
6. Discussion section has not been well developed. It should be supported with related theories and literature
|
We thank the reviewer for this content. We have added 3 sections, generalizability, limitations and future directions one on Limitations. We have also added some content on previous literature, and how it relates to the content and expanded our conclusions
Based on previous studies, these modifications likely protect against falls and ensure enhanced livability and quality of life. Previous analyses of Medicare beneficiaries have demonstrated that bathroom modifications prevent repeated falls, and that racial and ethnic minorities are less likely to have bathroom modifications, leading to subsequent falls. But these barriers, particularly for low income and racial-ethnic minorities also exist at the and community societal levels. This study encourages a paradigm shift in the clinical and healthcare space when operationalizing AIP. In the clinical literature, AIP is operationalized as primarily resolving or managing disabilities in the homes, focusing on The individual level of the physical/built domain. While essential, this is reductive. Successful AIP should address health outcomes expansively. This includes trying to positively influence levels such as the community (to improve neighborhood livability and function) and Societal levels (such discrimination). Often, influencing these levels is outside the immediate scope of the healthcare practitioners. Where clinical practitioners assess they cannot be impactful, they should seek a community partnership to support impacting more levels of influence. Partnering with community organizations is feasible and necessary for addressing all levels of health outcomes. . The study provides conceptual context for future studies which could follow participants longitudinally to gauge effectiveness on the individual, interpersonal, community and societal effects. Occupational therapists, aging in place specialists, and other clinical providers and interventions can be co-designed with community partners to minimize challenges to completing ADLs and IADLs, and also to support long term goals. These goals can support community health, such as intergenerational home ownership, neighborhood livability, and reduce the burden of cash repairs. It also provides an avenue for a new conceptualization of the idea of AIP in the clinical space. In partnership, providers and community based organizations are able to influence more levels of health than if working in silo. Access to housing is not accessible housing, and to remove barriers, practitioners and community based organizations should expand their appreciation of obstacles to include historical, contemporary, economic, and environmental factors to work toward equity in aging in place for all.
The limitations of the data provides a backdrop for the discussion of (re)conceptualizing AIP. From an implementation standpoint, clinical interventions have primarily focused on AIP as the intersection of Physical/Built environment and the individual and interpersonal levels. However AIP includes community levels (such as neighborhood livability) and societal domains (such as healthcare policy, discrimination and social norms). RT-R concentrates work in particular neighborhoods to support individual health, and as a way to improve neighborhood livability Partnership with CBOs and healthcare may more effectively address AIP by impacting more levels of influence. Proactive partnerships between CBOs and healthcare organizations also create potential for more AIP research, because outcomes can be robustly linked to health outcomes. Future studies which could follow participants longitudinally to gauge effectiveness on the individual, interpersonal, community and societal effects. Occupational therapists, aging in place specialists, and other clinical providers and interventions can be co-designed with community partners to minimize challenges to completing ADLs and IADLs, and also to support long term goals. These goals can support community health, such as intergenerational home ownership, neighborhood livability, and reduce the burden of cash repairs. It also provides an avenue for a new conceptualization of the idea of AIP in the clinical space
This study also serves as a call to action for healthcare practitioners and educators to change how we define and support AIP. Focusing on the physical and built environment at the individual levels (i.e. “provid[ing] tools to allow older adults to stay in their homes as they age by supporting the necessary modifications” Entrenches inequities by not addressing community and societal barriers.
|
7. Please provide limitations of the study
|
Thank you, A limitation section has been added There are several limitations within the study. RT-R collects information relevant to the domain of physical and built environment, data points relevant to the health domain (such improvement in IADLs, or reduction in falls are not collected by the RT-R program. This study is cross sectional, not longitudinal, and therefore we are unable to gauge the long term impact of the modifications. It is likely that repairs, such as installing toilets and repairing hand railings have supported AIP for participants. Analytically there were limited statistical tests appropriate to this data. ANOVA to detect significance between categories of homeowners was not appropriate due to the clustering (instances of multiple occupants in one house). Analysis comparing different types of repairs at the home level was also not appropriate, because homes received multiple repairs in multiple categories, violating independence of assumptions. The small sample size, (n=33 homes, and n=47 participants) made a mixed effect regression inappropriate, due to being underpowered. All data was collected in a single zip code, generalizability of the findings are limited.
|
8. Please provide implications of the study |
Thank you reviewer, an implications section has been added.
There are several implications for this study. Most significantly is that the concept of the “AIP” paradigm which primarily focuses on the physical structure of the home may unintentionally be exclusionary to diverse populations. AIP, as it is defined by the National Institute of Aging and the AARP, does not provide options for individuals who do not have agency over their living domain, nor does it address livability factors which are relevant to the immediate environment outside the dwelling. In the context of health, community and social factors have significant implications for health, and have historically been inequitable for racial and ethnic minorities. From an international standpoint, the majority of AIP literature is based in the United States, or western Europe; this is a gap. Audiences outside of these countries should study and disseminate culturally relevant definitions of AIP if they already exist; if a country does not currently have infrastructure or a paradigm for AIP, they could consider building infrastructure which includes all of the levels health and look for more comprehensive ways to define, and share best practices on AIP. Prior research has been performed with community based research partnerships (CBRP) in order to achieve health equity. Ward and colleagues (2018) posited with a Conceptual Model for Evaluating Equity within the Context of CBPR Partnerships framework that there were various long-term outcome measures to come from CBRP effectiveness in creating equity, which would include an improvement in social and environmental conditions directly within inequitable communities [37]. Additionally, improvements within individuals’ physical, mental and social health could be addressed within communities which had health inequities with the use of CBRPs. Limited research literature to date has addressed research participation with community based organizations; however, this study provides early foundational knowledge of some of the manners in which outputs from a CBRP could add individual and community level value to better address rectifying health inequities.
|
Reviewer 2 Report
Comments and Suggestions for Authors
Dear Editors / Dear Author(s),
Thank you for the opportunity to review the manuscript “Aging in Place and Healthcare Equity: How Rebuilding Together Supports Individuals Living Longer in Their Homes and Their Communities. “
The article explores an essential topic related to aging in place (AIP), health equity, and the role of community-based organizations in fostering aging-friendly environments. I appreciate the author’s critical approach and how the analysis is anchored in a solid conceptual framework.
Below, I present my evaluation, emphasizing the manuscript's strengths and providing constructive suggestions for improvement.
The manuscript addresses a critical issue related to policy and practice in the field of aging and public health. Exploring how structural, economic, and racial inequalities influence the capacity to age in place is both relevant and essential. Furthermore, the use of the National Institute on Minority Health and Health Disparities (NIMHD) Research Framework offers a compelling multi-level perspective to analyze disparities in aging in place. Combining a community-based case study (Rebuilding Together Richmond) with theoretical insights adds depth and practical significance to the paper.
I suggest four areas of improvement:
Methodology
While the descriptive analysis is informative, the methodology section would benefit from more significant detail. I encourage the authors to clarify the study design (e.g., observational, retrospective, case study), the criteria for inclusion and categorization of repairs, and how classifications (e.g., “occupational” vs. “structural”) were derived and validated.
Generalizability of the Results
The study focuses on a single zip code and has a relatively small sample size. The manuscript would be enhanced by a clearer discussion of the limitations and the contextual specificity of the findings.
Data Interpretation
Further analytical depth could enhance the results section, particularly concerning the relationships between demographic characteristics and types of repairs. The discussion could better link the descriptive findings to the existing literature and broader debates on health equity and aging policy.
Style and Conciseness
I suggest clarifying from the beginning what the rebuilding together intervention is. It seems to be a pivotal concept (as it is also included in Table 1), but it is not defined.
Figure 1 introduces and briefly discusses the framework, but its presence in the rest of the article is limited. The discussion and results sections do not consistently refer back to the framework or use it as a tool to interpret findings in a structured manner. After its initial appearance, Figure 1 is not referenced in depth or critically analyzed. It functions more as a backdrop than as a truly operational analytical framework.
As shown in Table 1, while it tries to align clinical and community interventions with levels of influence, the remainder of the article does not systematically analyze the empirical data (from Rebuilding Together Richmond) according to the NIMHD levels and domains.
Certain sections—especially the introduction and discussion—have repetitive elements. A more concise narrative structure would enhance readability without compromising depth.
I suggest moderate revisions. The topic is engaging, and the contribution is clear, but adjustments are necessary to enhance methodological transparency and analytical depth.
Author Response
Reviewer 2
Reviewer Comment |
Author Response |
Thank you for the opportunity to review the manuscript “Aging in Place and Healthcare Equity: How Rebuilding Together Supports Individuals Living Longer in Their Homes and Their Communities. “
The article explores an essential topic related to aging in place (AIP), health equity, and the role of community-based organizations in fostering aging-friendly environments. I appreciate the author’s critical approach and how the analysis is anchored in a solid conceptual framework.
Below, I present my evaluation, emphasizing the manuscript's strengths and providing constructive suggestions for improvement.
The manuscript addresses a critical issue related to policy and practice in the field of aging and public health. Exploring how structural, economic, and racial inequalities influence the capacity to age in place is both relevant and essential. Furthermore, the use of the National Institute on Minority Health and Health Disparities (NIMHD) Research Framework offers a compelling multi-level perspective to analyze disparities in aging in place. Combining a community-based case study (Rebuilding Together Richmond) with theoretical insights adds depth and practical significance to the paper. |
We would like to thank the reviewer for the comments. |
Methodology: |
We Thank the reviewer for the comment, WIthin the methods sections, the sentence has now been updated to: To examine the impact of AIP ecology across all levels of influence and study the structural impact and outcomes for one community, retrospective secondary data from Rebuilding Together Richmond (RT-R) were analyzed. We have also clarified study dates.
We have also included the rationale for limited statistical analysis in our limitation section. Analytically there were limited statistical tests appropriate to this data. ANOVA to detect significance between categories of homeowners was not appropriate due to the clustering (instances of multiple occupants in one house). Analysis comparing different types of repairs at the home level was also not appropriate, because homes received multiple repairs in multiple categories, violating independence of assumptions. The small sample size, (n=33 homes, and n=47 participants) made a mixed effect regression inappropriate, due to being underpowered. All data was collected in a single zip code, generalizability of the findings are limited. |
The criteria for inclusion and categorization of repairs, |
Thank you reviewer, we have clarified we used the Rebuilding together categorization in the methods section “ Categorization of repairs was based on the RT-R categorization data dictionary” |
and how classifications (e.g., “occupational” vs. “structural”) were derived and validated. |
Further details and organization has been made in the Materials/Methods section as follows to better elucidate the classification system. “RT-R uses a 22-category system to organize repairs. Classification of repairs was based on the RT-R categorization data dictionary (provided to the research team by RT-R) with two research team members analyzing and verifying each repair. The 22 repair categories included: attic, bathroom, bedroom, crawl, dining, electrical, external door, fence, gutters, hall, HVAC, kitchen, landscape, and miscellaneous—paint, pests, plumbing, porch/deck, ramp, roof, siding, windows. Recipients of repairs were categorized into four groups: Aging with a disability (65+ and a disability, Aging in Place (65 + and no disability), Living with a disability (Under 65 with a disability), and all others. To further delineate the classification of home repairs, each individual repair was classified as either “structural” or “occupational” based on dual reviewer alignment with most homes receiving multiple repairs. The American Occupational Therapy Association (AOTA) Occupational Therapy Practice Framework served as a basis for item categorization and determination. (Table 2 and Table 3). Structural repairs were classified as repairs that addressed the home’s safety and/or aesthetics. Some examples of these repairs included the replacement of the outlet cover, new paint, and replacement of the window(s). Occupational repairs were individually classified based on the potential impact on a resident’s functional mobility, activities of daily living (ADLs), and/or instrumental activities of daily living (IADLs). Some examples of occupational repairs included the addition and/or replacement of handrails for stairs, raising toilet heights, and the addition of grab bars within the shower, amongst other home repairs. This separation allows for an analysis of how rehabilitation practitioners may impart value in the remodeling process while in no way diminishing the value brought to the home from structural repairs in terms of additional home safety and economic value. “ |
Generalizability of the Results: The study focuses on a single zip code and has a relatively small sample size. The manuscript would be enhanced by a clearer discussion of the limitations and the contextual specificity of the findings. |
Thank you reviewer, we have added a limitations section to address that these findings have generalizability limitations: Limitations There are several limitations within the study. Analytically, it is underpowered due to the small sample size, (n=33 homes, and n=47 participants). There is also a lack of independence of observations, as several homes received multiple types of repairs (i.e. structural and occupational). The data were limited, only including basic demographics and types of repairs performed. The sample frame is of a single zip code, because RT-R uses home modifications as a way to support not only individual and interpersonal health, but as a way to improve neighborhood livability. The generalizability of the analytic findings are limited. This study is cross sectional, not longitudinal, and therefore we are unable to gauge the long term impact of the modifications..
|
Data Interpretation |
Thank you reviewer, we agree. We have discussed in detail why further statistical testing would have inappropriate, however we have added context throughout the entire paper about access to healthcare and health policy, and clarified our paper is an invitation to reimagine Aging in Place. |
Style and Conciseness |
Additional information has now been added in the Introduction section to detail Rebuilding Together earlier in the paper. The new second paragraph of the Introduction now reads “ This study bridges the gap between theoretical aging-in-place literature and the clinical practice of mediating aging-in-place barriers through a collaboration with the organization, Rebuilding Together. Rebuilding Together, a nonprofit organization who performs home repairs for individuals in need, allowed for a unique understanding through the use of descriptive analyses of the individual needs to enable sustained livelihood in home environments. “ |
Figure 1 introduces and briefly discusses the framework, but its presence in the rest of the article is limited. The discussion and results sections do not consistently refer back to the framework or use it as a tool to interpret findings in a structured manner. After its initial appearance, Figure 1 is not referenced in depth or critically analyzed. It functions more as a backdrop than as a truly operational analytical framework. |
We appreciate this comment from the reviewer, we now consistently refer to the framework throughout our manuscript, including identifying the levels of influence. |
As shown in Table 1, while it tries to align clinical and community interventions with levels of influence, the remainder of the article does not systematically analyze the empirical data (from Rebuilding Together Richmond) according to the NIMHD levels and domains. |
We appreciate the reviewer's feedback. We align the levels of influence to the actions of health and systems partners more consistently throughout the paper. |
Certain sections—especially the introduction and discussion—have repetitive elements. A more concise narrative structure would enhance readability without compromising depth. |
Thank you reviewer, we agree and have removed repetitive elements while focusing on readability |
Reviewer 3 Report
Comments and Suggestions for Authors
This study identifies the importance of housing stability and access to healthcare for successful Aging in Place (AIP), and the benefits that come with AIP relative to institutionalization of the aging. It also identifies obstacles and housing disparities that make AIP inaccessible for some, with greater impact on racial and ethnic minorities and those with disabilities. It calls for integrating clinical AIP interventions with community based solutions, drawing together physical environments (with structural and occupational repairs) and healthcare systems. This paper would be strengthened by inclusion of case studies, interviews, or additional examples of how to achieve the partnerships and integration of services that will provide improved AIP outcomes from a health equity perspective.
Author Response
Reviewer 3
Reviewer Comment |
Authors Response |
This paper would be strengthened by inclusion of case studies, interviews, or additional examples of how to achieve the partnerships and integration of services that will provide improved AIP outcomes from a health equity perspective. |
Thank you reviewer, we agree however As we were exempt in our IRB, we were unable to perform interviews or have access to case studies from this research. We were able to include additional examples of how partnerships could be achieved to attempt to improve AIP outcomes. |
Round 2
Reviewer 1 Report
Comments and Suggestions for Authors
I would like to thank the authors for considering the comments and changing the manuscript accordingly.
Author Response
Thank you for your kind comments.
Reviewer 3 Report
Comments and Suggestions for Authors
A future study of this topic could benefit from inclusion of case studies and interviews.
This manuscript can be published with minor editorial revisions in the abstract. Several terms are hyphenated in the text and should be corrected: modifications, analyzed, categories, address.
Author Response
Response:
This statement is true in the abstract contained within the mdpi website. I did not see any way in which to delete these hyphens as they are not contained within the manuscript itself. In order to ensure there were no hyphenations within the abstract (if it was pulled directly from the document), additional spaces were created to ensure that no text would wrap around to the next line and therefore potentially create a hyphen. Hyphenations were added within the document to ensure consistency with the term community-based organizations. The author citation along the side of the front page does not allow for full authorship and manuscript citation without shifting text onto the next page, so we will defer to the journal to update the full citation.
I have attached the document for review. Please let us know if you have any other questions or concerns. Thank you.
